# Soluble Urokinase-Type Plasminogen Activator Receptor (suPAR) and Growth Differentiation Factor-15 (GDF-15) Levels Are Significantly Associated with Endothelial Injury Indices in Adult Allogeneic Hematopoietic Cell Transplantation Recipients

**DOI:** 10.3390/ijms25010231

**Published:** 2023-12-23

**Authors:** Eleni Gavriilaki, Zoi Bousiou, Ioannis Batsis, Anna Vardi, Despina Mallouri, Evaggelia-Evdoxia Koravou, Georgia Konstantinidou, Nikolaos Spyridis, Georgios Karavalakis, Foteini Noli, Vasileios Patriarcheas, Marianna Masmanidou, Tasoula Touloumenidou, Apostolia Papalexandri, Christos Poziopoulos, Evangelia Yannaki, Ioanna Sakellari, Marianna Politou, Ioannis Papassotiriou

**Affiliations:** 1Second Propedeutic Department of Internal Medicine, Hippocration Hospital, Aristotle University of Thessaloniki, 54642 Thessaloniki, Greece; 2BMT Unit, Hematology Department, George Papanicolaou General Hospital, 57010 Thessaloniki, Greece; boussiou_z@hotmail.com (Z.B.); mpatsisioannis.gpapanikolaou@n3.syzefxis.gov.gr (I.B.); anna_vardi@yahoo.com (A.V.); dmallouri@gmail.com (D.M.); evakikor@gmail.com (E.-E.K.); konstakg@yahoo.gr (G.K.); spyridisnik@hotmail.com (N.S.); giorgos.karavalakis@gmail.com (G.K.); fnol25@otenet.gr (F.N.); vpatriar@gmail.com (V.P.); marianareti@gmail.com (M.M.); tasoula.touloumenidou@gmail.com (T.T.); lila.papalexandri@gmail.com (A.P.); eyannaki@uw.edu (E.Y.); bmt@papanikolaou.gr (I.S.); 3Department of Hematology, Metropolitan Hospital, Neo Faliro, 18547 Athens, Greece; cpozi@otenet.gr; 4Hematology Laboratory-Blood Bank, Aretaieion Hospital, School of Medicine, National and Kapodistrian University of Athens, 11527 Athens, Greece; mpolitou@med.uoa.gr; 5First Department of Pediatrics, School of Medicine, National and Kapodistrian University of Athens, 15772 Athens, Greece; ipapassotiriou@gmail.com

**Keywords:** soluble urokinase plasminogen activator receptor (suPAR), growth differentiation factor-15 (GDF-15), endothelial dysfunction, allogeneic hematopoietic stem cell transplantation, HSCT-TMA, GvHD

## Abstract

Hematopoietic stem cell transplantation-associated thrombotic microangiopathy (HSCT-TMA) and graft-versus-host disease (GvHD) represent life-threatening syndromes after allogeneic hematopoietic stem cell transplantation (allo-HSCT). In both conditions, endothelial dysfunction is a common denominator, and development of relevant biomarkers is of high importance for both diagnosis and prognosis. Despite the fact that soluble urokinase plasminogen activator receptor (suPAR) and growth differentiation factor-15 (GDF-15) have been determined as endothelial injury indices in various clinical settings, their role in HSCT-related complications remains unexplored. In this context, we used immunoenzymatic methods to measure suPAR and GDF-15 levels in HSCT-TMA, acute and/or chronic GVHD, control HSCT recipients, and apparently healthy individuals of similar age and gender. We found considerably greater SuPAR and GDF-15 levels in HSCT-TMA and GVHD patients compared to allo-HSCT and healthy patients. Both GDF-15 and suPAR concentrations were linked to EASIX at day 100 and last follow-up. SuPAR was associated with creatinine and platelets at day 100 and last follow-up, while GDF-15 was associated only with platelets, suggesting that laboratory values do not drive EASIX. SuPAR, but not GDF-15, was related to soluble C5b-9 levels, a sign of increased HSCT-TMA risk. Our study shows for the first time that suPAR and GDF-15 indicate endothelial damage in allo-HSCT recipients. Rigorous validation of these biomarkers in many cohorts may provide utility for their usefulness in identifying and stratifying allo-HSCT recipients with endothelial cell impairment.

## 1. Introduction

The vascular endothelium is a semipermeable cellular monolayer that coats the interior walls of arteries, capillaries, and veins and separates circulating blood from tissues [1]. The vascular endothelium not only serves as a simple barrier but also plays an important role in the maintenance of homeostasis and is involved in several important functions, including the regulation of vascular tone, hemostasis, inflammation, and exchange between blood and tissue compartments [2]. Under normal circumstances, there is an equilibrium between procoagulant and anticoagulant pathways, which enables a prompt response to injury when it is needed. The impact of allogeneic hematopoietic cell transplantation (allo-HSCT) on endothelial cells has been widely recognized, with evidence indicating that these cells undergo various alterations prior to, during, and following the procedure. As a result, endothelial injury occurs, giving rise to severe consequences that pose a threat to the patient’s life [3]. Endothelial dysfunction is a pivotal factor in the pathogenesis of transplant-related complications, which are commonly termed endothelial injury syndromes that include hematopoietic stem cell transplantation-associated thrombotic microangiopathy (HSCT-TMA), graft-versus-host disease (GvHD), sinusoidal obstruction syndrome (SOS), capillary leak syndrome, engraftment syndrome, diffuse alveolar hemorrhage (DAH), and idiopathic pneumonia syndrome (IPS) [4,5]. HSCT-TMA constitutes a well-documented complication of allo-HSCT associated with increased morbidity and mortality. HSCT-TMA is an increasingly recognized, highly heterogenous, and systemic clinical entity that encompasses features of thrombotic microangiopathies, including microangiopathic hemolytic anemia, platelet consumption, peripheral blood schistocytes, and organ dysfunction [6]. The understanding of pathophysiology has rapidly evolved over the last few years. Elevated concentrations of soluble C5b-9 (sC5b-9) have been suggested as a potential biomarker for complement activation and a negative prognostic factor in patients with HSCT-TMA [7].

A diagnosis of HSCT-TMA is established when four or more of the subsequent seven features manifest twice within a span of fourteen days: anemia, defined as failure to achieve transfusion independence despite neutrophil engraftment; hemoglobin decline by ≥1 g/dL or new-onset transfusion dependence; thrombocytopenia, defined as failure to achieve platelet engraftment, higher-than-expected transfusion needs, refractory to platelet transfusions, or ≥50% reduction in baseline platelet count after full platelet engraftment; lactate dehydrogenase (LDH) exceeding the upper limit of normal (ULN); schistocytes; hypertension; soluble C5b-9 (sC5b-9) exceeding the ULN; and proteinuria (≥1 mg/mg random urine protein-to-creatinine ratio (rUPCR)) [8].

Considering the pivotal role of endothelial dysfunction in the pathogenesis of potentially lethal complications of allo-HSCT, the development of biomarkers that reflect the course of endothelial dysfunction and could serve as prognostic tools for endothelial injury was a sine qua non. The Endothelial Activation and Stress Index (EASIX) score is based on the calculation of three simple laboratory parameters: lactate dehydrogenase (LDH) multiplied by creatinine, the product of which is divided by platelet count, was established as a prognostic marker and as a surrogate indicator of endothelial dysfunction [9]. Several studies have shown that the EASIX score can accurately predict endothelial complications following allo-HSCT [10,11]. Moreover, it has been recently shown that EASIX can serve as a predictive marker for both complement activation and overall survival in patients with GvHD and HSCT-TMA [12]. soluble urokinase plasminogen activator receptor (suPAR) and growth differentiation factor-15 (GDF-15) have been determined as endothelial injury indices in various clinical settings in both hematological and non-hematological diseases [13,14,15,16].

Therefore, the aim of the study was to assess whether suPAR and GDF-15 reflect endothelial injury in allo-HSCT recipients as well as the possible correlation of these biomarkers with EASIX.

## 2. Results

### 2.1. Patient Characteristics

We studied 20 HSCT-TMA, 20 GVHD, 20 control allo-HSCT patients, and 20 healthy controls. The study group characteristics are presented in Table 1. HSC-TMA developed at a median of 125 post-transplant days (range 9–2931), whereas the first day of confirmed GVHD diagnosis was at a median of 78 post-transplant days (range 16–145).

### 2.2. suPAR and GDF-15 Levels 

We found significantly higher suPAR and GDF-15 levels in HSCT-TMA and GVHD patients compared to allo-HSCT and healthy controls (*p* < 0.001, Bonferroni’s correction) (Figure 1). Then, we further analyzed characteristics of the allo-HSCT population.

### 2.3. Correlation with EASIX

SuPAR and GDF-15 concentrations were found to be correlated with EASIX at day 100 (r = 0.351, *p* = 0.012 and r = 0.338, *p* = 0.015, respectively) and at the last follow-up (r = 0.473, *p* < 0.001 and r = 0.410, *p* = 0.020, respectively). Among the laboratory values utilized in the calculation of EASIX—namely LDH, creatinine, and platelets—suPAR was associated with creatinine and platelets at day 100 and last follow-up, while GDF-15 was associated only with platelets at both time points, suggesting that the association with EASIX is not driven by laboratory values per se. It is noteworthy that only suPAR levels, and not GDF-15 levels, exhibited an association with soluble C5b-9 levels (*p* = 0.013) (Figure 2). This association serves as an indicator of increased risk in thrombotic microangiopathy linked with HSCT-TMA.

## 3. Discussion

Two decades ago, HSCT-TMA was considered a form of thrombotic thrombocytopenic purpura (TTP), and plasmapheresis was the first-line therapeutic approach [17]. However, ADAMTS13 was not found to be deficient in HSCT-TMA patients, and plasmapheresis was inefficient in most cases [18,19]. Comprehension of the pathogenesis of Complement-Mediated Hemolytic Uremic Syndrome (CM-HUS) in combination with accumulating genetic and functional evidence of complement activation in the pathogenesis of HSCT-TMA made clinicians draw the conclusion that HSCT-TMA resembles CM-HUS [20,21]. In the realm of pathophysiology, Luft et al. have recently proposed a three-hit theory to describe HSCT-TMA pathogenesis [3]. This consists of the following: (1) An endothelial vulnerability, which refers to pre-existing endothelial injury or underlying predisposition to complement activation, represents the first hit. This suggests that pre-existing defects have the potential to influence the endothelial response and/or to supplement subsequent challenges that may arise following allo-HSCT. This conclusion is drawn from the finding that adverse outcomes are linked to particular single nucleotide polymorphisms (SNPs) in recipient genes or elevated plasma markers that indicate endothelial cell distress before allo-HSCT [22]. (2) The occurrence of a harmful event in the vascular endothelium, such as the conditioning regimen, triggers the release of proinflammatory cytokines and procoagulant proteins while also depleting protective mechanisms like nitric oxide and vascular endothelial growth factor. These factors contribute to endothelial injury and initiate the complement cascade, representing the second hit. Furthermore, the activation of the complement system is exacerbated by the presence of additional insults, such as infections or the administration of drugs or immunosuppressants. This heightened complement activation ultimately results in the production of microthrombi, as indicated by hit 3. This well-established hyperactivation of the complement system is also supported by research studies which have shown mounting evidence that inhibiting the C5 level of the complement system with eculizumab—approved for use in aHUS—is effective in many HSCT-TMA patients, leading to better overall response and survival rates [23,24]. However, today, there is no approved targeted therapy available, and there are several ongoing trials investigating complement inhibitors [25,26,27,28] as potential treatments in both adult and pediatric patients with HSCT-TMA.

Graft-versus-host disease (GvHD) is a significant complication and one of the leading causes of morbidity and mortality after allogeneic HCT. GvHD arises when immunocompetent T cells attack and destroy host tissues whose antigens differ from the donor’s due to incompatibility, thus rejecting the transplanted cells in the recipient [29]. In the past, GvHD was classified as acute when the onset occurred before 100 days and as chronic when the onset occurred after 100 days of stem cell infusion, respectively. The 2005 National Institute of Health (NIH) Consensus Conference proposed new criteria for diagnosis and classification of GvHD, which should be carried out based on clinical manifestation and not the onset time after stem cell infusion [30]. In acute GvHD, the skin, the liver, and the gastrointestinal tract are the principal target organs. Patients presenting with clinical symptoms of acute GvHD within 100 days are considered to have “classic acute GvHD”, whereas patients with symptoms after 100 days are classified as “late-onset GvHD”. Acute GvHD can also be further classified into mild, moderate, severe, and very severe based on several grading systems that consider the affected organs’ involvement. Despite progression in prophylaxis, disease burden remains high, and acute GvHD is associated with higher readmission rates and increased mortality [31]. In GvHD, the current treatment approach relies on disease severity. For low-grade (i.e., grades 1 and 2) isolated skin GvHD, topical steroids are the choice of treatment, whereas for more severe skin GvHD or gastrointestinal and liver GvHD, systemic steroids are the first-line treatment [32,33]. If treatment with steroids fails (steroid-refractory GvHD), then patients should be started on ruxolitinib, a JAK1/2 inhibitor which suppresses inflammatory signaling by cytokines involved in GvHD pathophysiology [34]. Even in patients treated with ruxolitinib, approximately one-third of them fail to attain a complete or partial response, and for this reason, there is ongoing research for third-line treatment options [35]. On the other hand, chronic GvHD constitutes a distinct clinical entity with a discrete pathophysiology. In patients with chronic GvHD, any body organ can be affected, leading to a great heterogeneity of signs and symptoms. The risk factors that have been identified and established include the utilization of growth factor-mobilized peripheral blood stem cells, the transplantation of grafts from mismatched or unrelated donors, the transplantation from female to male recipients, advanced recipient age, and a history of acute GvHD [36].

Endothelial dysfunction is a common denominator for both HSCT-TMA and GvHD. It is well known that the cornerstone of HSCT-TMA pathogenesis is endothelial dysfunction in combination, which initiates complement activation, leading to the development of thrombi enriched in platelets within the microvasculature [37]. Recent studies have shown the potential involvement of endothelial dysfunction in the pathogenesis of GvHD, postulating that endothelial dysfunction is a shared characteristic of both diseases. Interestingly, the provided evidence regarding the potential involvement of endothelial dysfunction in the pathogenesis of GvHD and the results from these studies suggest that endothelial cells may be vulnerable to assault by alloreactive T lymphocytes originating from the donor [38,39,40,41]. These findings are consistent with the association of many endothelium-related variables with GVHD outcomes [9,42]. Regarding the complement system, its activation has been extensively documented in HSCT-TMA, whereas it seems that it is not involved in GvHD [43]. 

Urokinase plasminogen activator (uPAR) is present on the cell surface membrane and plays a crucial role in the activation of plasminogen in the process of fibrinolysis as well as in other essential cellular activities such as proliferation, adhesion, angiogenesis, and inflammation. It is primarily expressed in immune cells, endothelial cells, and cancer cells [44]. The soluble version of the receptor uPAR is known as soluble uPAR (suPAR) and arises after the proteolytic cleavage of uPAR. The suPAR is present in various bodily fluids, including blood plasma, serum, cerebrospinal fluid, saliva, and urine [45]. It has been demonstrated that in the presence of inflammatory stimuli, immune cells release higher amounts of suPAR; thus, it is believed that the concentration of suPAR in the bloodstream reflects an individual’s degree of inflammation and immunological activation, regardless of the underlying cause. Elevated suPAR levels are seen in several diseases, such as cardiovascular disease (CVD), renal disease, diabetes mellitus, malignancies, rheumatic diseases, and infections [13,14,46]. Elevated suPAR levels are associated with endothelial dysfunction and atherosclerosis in patients with CVD, and suPAR possesses the potential to be a prognostic indicator in individuals diagnosed with CVD [47,48]. Apart from CVD, it has been shown that elevated levels of suPAR have a direct impact on renal function through the pathological activation of αvβ3 integrin found in podocytes [49]. In AL amyloidosis patients, suPAR levels predict renal outcomes, particularly in cases where there is progression to end-stage renal disease (ESRD) necessitating dialysis [50]. Moreover, in a recent study by Lafon et al., suPAR appeared to be a promising tool for predicting the deterioration of patients with suspected bacterial acute infection upon admission to the emergency department (ED) [51]. Despite its implication in several diseases, the role of suPAR in allogeneic HCT, is less studied. In a single-center study by Haastrup and colleagues in 2011 [52], notable elevations in suPAR levels during the pretransplant conditioning phase in allo-HSCT were illustrated. This observation could be explained by the fact that conditioning regimens elicit tissue damage and stimulate the release of inflammatory cytokines, providing further propagation of the inflammatory response and upstream expression of UPAR and suPAR release [53].

Growth differentiation factor-15 (GDF-15), initially termed macrophage inhibitory factor-1 (MIC-1) in 1997 by Bootcov et al. [54], is a cytokine belonging to the transforming growth factor beta (TGF-ß) family. GDF-15a has little to no constitutive expression outside of reproductive organs, liver, and kidneys; nevertheless, its expression can be induced in numerous cell types under stress circumstances [55]. The exact function of GDF-15 is still not completely understood; however, it appears to be involved in several cellular processes such as apoptosis, cell repair, and cell development and has a role in the regulation of inflammatory pathways. GDF15 serves as a broad-spectrum biomarker for several diseases, such as cardiovascular, pulmonary, and renal as well as for malignancies [15]. It has been shown that GDF-15 can modify vascular relaxation and contraction responses in an endothelial-dependent manner that involves the nitric oxide pathway [56]. Endothelial dysfunction in lesion-prone regions of the arterial vasculature is well established to contribute considerably to the pathobiology of atherosclerotic cardiovascular disease, and GDF-15 has been found to be abundant in atherosclerotic plaques in CVD patients [57]. GDF-15 also plays a crucial role in the inhibition of hepcidin secretion and subsequent tissue iron overload observed in patients with disorders characterized by inadequate erythropoiesis [58]. GDF-15 levels have been found to be elevated in patients with SCD, thalassemia (both major and intermediate), and the compound heterozygotes HbS/βthal and to correlate with endothelial dysfunction and atherosclerosis in the transfusion-dependent form (tdt) beta-thalassemia [59,60]. Despite its correlation with endothelial injury, the precise functional role of GDF15 in endothelial cells remains mainly unclear [61]. Both in vitro and in vivo, GDF15 seems to provide protection against tissue damage through its anti-inflammatory and antiapoptotic capabilities [58]. Moreover, GDF-15 regulates signaling transduction pathways on endothelial cells that are essential for proliferation and angiogenesis through activation ALK activin receptor like kinase (ALK) receptors [62].

Notwithstanding the correlation of suPAR and GDF-15 levels with endothelial dysfunction in several clinical entities, these biomarkers have never been studied in patients with HSCT-TMA and GVHD. To date, there is a lack of standardized laboratory parameters that specifically address endothelial dysfunction in these patients. The EASIX score, which is based on the calculation of LDH multiplied by Cr, divided by platelet count, has emerged as a potential prognostic biomarker of endothelial dysfunction in patients with HSCT-TMA and GVHD [63,64]. 

In our study, we found significantly higher suPAR and GDF-15 levels in HSCT-TMA and GVHD patients compared to allo-HSCT and healthy controls. Moreover, both GDF-15 and suPAR concentrations were associated with EASIX at day 100 and the last follow-up, and these findings are comparable with our previous study where EASIX at the same time points was significantly higher in HSCT-TMA and GVHD compared with controls [63]. Interestingly, suPAR was associated with creatinine and platelets at day 100 and last follow-up, while GDF-15 was only associated with platelets at both time points. These findings indicate that the association between EASIX and these laboratory values is not solely influenced by the values themselves. It is worth noting that the association between suPAR levels and soluble C5b-9 levels, which serve as a marker for high risk in HSCT-TMA, was statistically significant, but no such association was observed with GDF-15 levels. In our single-center study, we demonstrated that elevated levels of suPAR and GDF-15 reflect endothelial injury in HSCT-TMA and GVHD patients after allo-HSCT. Consistent with observations in several patient cohorts, suPAR has been identified as a biomarker for renal dysfunction, specifically indicating a heightened risk in syndromes associated with endothelial injury, with a particular focus on HSCT-TMA. The observed associations of these biomarkers in allo-HSCT recipients align with their roles in other diseases. As proposed by the three-hit hypothesis, endothelial injury will lead to the release of proinflammatory cytokines, procoagulant proteins, and depletion of protective mechanisms like nitric oxide [65]. Moreover, pathologic complement activation contributes further to endothelial injury and leads to platelet activation, small vessel thrombosis, and microangiopathic hemolytic anemia [66].

However, our study is subject to various limitations. First of all, we conducted a single-center study, involving a limited number of patients; consequently, larger cohorts are required to validate our findings. Due to the limited number of patients included in our investigation, we encountered limitations in evaluating the impact of distinct conditioning regimens or specific disease cohorts on endothelial function. However, we believe that the sample size adequately represents the diversity within these patient populations, considering the potential variability in responses post HSCT among different individuals since the incidences of post-HSCT complications in our department align with reported incidences across the globe. Bias was minimized by ensuring that 20 patients were consecutively enrolled from each group. Patients with additional endothelial dysfunction syndromes, such as sinusoidal obstructive syndrome/veno-occlusive disease, were not included in our study due to the infrequent occurrence of this condition among our patient population.

In conclusion, our study shows for the first time that suPAR and GDF-15 levels reflect endothelial injury in allo-HSCT patients. Hematopoietic stem cell transplantation is linked to notable endothelial activation, often leading to severe post-transplant complications associated with endothelial dysfunction. The latter constitutes a common denominator in both HSCT-TMA and GvHD. Hence, the identification of endothelium biomarkers for prompt diagnosis and prognosis of endothelial dysfunction in allo-HSCT recipients could potentially facilitate early intervention and mitigate the risk of irreversible endothelial cell impairment. Integration of these biomarkers into clinical trials could lead to their subsequent use in daily clinical practice for both diagnostic and prognostic purposes.

## 4. Materials and Methods

### 4.1. Patients and Study Design

In this case–control study, we recruited in a 1:1:1:1 ratio consecutive adult patients diagnosed with HSCT-TMA, patients diagnosed with acute and/or chronic GvHD, control allo-HSCT recipients who did not have GvHD or HSCT-TMA, and apparently healthy individuals of similar age and gender. All patients underwent allo-HSCT at our facility, which is accredited by the European Society for Blood and Marrow Transplantation (EBMT) and the Joint Accreditation Committee of the International Society for Cellular Therapy (ISCT), between January 2015 and June 2018. The institutional review board of G. Papanikolaou Hospital granted approval for this study, and written informed consent was obtained from all patients in adherence to the principles outlined in the Helsinki Declaration. Allo-HSCT was performed in adherence to the indications established by the European Society for Blood and Marrow Transplantation. A comprehensive initial assessment was performed on each patient, comprising the following: a meticulous physical examination; clinical determination of GvHD; if relevant, organ biopsy (with duodenum, terminal ileal, and colonic biopsies performed to exclude infection); evaluation of blood count; and assessment of biochemical values. In adherence to the established standard operating procedures of our institution, data were methodically gathered from our database. This included initial and subsequent demographic, clinical, and laboratory data from the patients’ plasma and sera samples were obtained from patients who had been diagnosed with GvHD or HSCT-TMA prior to the initiation of targeted therapy. Εthylenediaminetetraacetic acid (EDTA) samples were drawn. Regarding HSCT-TMA, the sampling day was the 125th post-transplant day (rage 9–2931), whereas the sampling day for GvHD was the 74th post-transplant day (range 12–141). Samples were collected from control subjects at the outpatient clinic during a prearranged appointment. Specimens were centrifuged, expeditiously aliquoted, and subsequently stored at a temperature of −80 °C.

### 4.2. GvHD Prophylaxis

The prophylactic protocol for graft-versus-host disease (GVHD) in myeloablative regimens, both from sibling and unrelated donors, involved the administration of a calcineurin inhibitor in combination with a short-term regimen of four post-transplant methotrexate doses. In the case of reduced conditioning regimens, the prophylactic regimen consisted of cyclosporine in conjunction with mycophenolate mofetil (MMF). The post-transplant prophylaxis against GVHD in haploidentical transplants consisted of cyclophosphamide (days 3 and 4) and cyclosporine plus MMF (day 5). The conditioning regimen for unrelated and haploidentical transplant recipients included the administration of anti-thymocyte globulin (ATG) at a dose of 5 mg/kg, as previously documented. Acute GVHD was evaluated and graded in accordance with the criteria proposed by Przepiorka et al. [67], whereas chronic GVHD was evaluated and graded in accordance with the criteria established by Sullivan and colleagues [56].

### 4.3. HSCT-TMA Diagnosis

HSCT-TMA diagnosis was established using the criteria outlined by the International Working Group (IWG): schistocytes greater than 4%, increased lactate dehydrogenase (LDH), thrombocytopenia, anemia, and decreased haptoglobin [68]. Following the identification of HSCT-TMA, the management considered all potential causative factors (including infections and calcineurin inhibitors) and implemented the following: cessation of calcineurin inhibitors, plasma infusion, plasma exchange, and corticosteroid administration. Upon retrospective reclassification of patients according to current consensus criteria [8], all patients were considered high-risk TMA.

### 4.4. Complement Activation Markers

To prevent repeated freeze–thaw cycles, aliquots of patients’ samples, including sera and EDTA plasma, were thawed solely on one occasion after being stored at a temperature of −80 °C. In this study, reagent cells resembling paroxysmal nocturnal hemoglobinuria (PNH) were subjected to incubation with sera, followed by exposure to the cell proliferation reagent 4-[3-(4-Iodophenyl)-2-(-4-nitrophenyl)-2H-5-tetrazolio]-1.3.-benzene disulfonate/WST-1 (Roche, Switzerland).. The absorbance was determined using an iMark Microplate Absorbance Reader (Bio-Rad, Hercules, CA, USA) at a wavelength of 490 nm, with a reference wavelength of 595 nm. A negative control was established using normal human serum, while a positive control was established using normal serum incubated with lipopolysaccharides. All samples underwent triple testing and were subsequently examined twice. In previous research, a positive test was operationally defined as the presence of nonviable cells exceeding a threshold of 20% [69]. 

### 4.5. EASIX, Soluble C5b-9/Membrane Attack Complex, suPAR, and GDF-15

The EASIX was calculated based on the proposed formula: lactate dehydrogenase (U/L) × creatinine (mg/dL)/thrombocytes (10^9^ cells per L). We calculated EASIX for every patient on days 0, 30, and 100 and at last follow-up. We measured, by means of immunoenzymatic techniques, the soluble C5b-9/membrane attack complex (Quidel, San Diego, CA, USA), suPAR (ViroGates A/S, Birkerod, Denmark), and GDF-15 (R&D Systems, Minneapolis, MN, USA).

### 4.6. Statistical Analysis

Data were subjected to analysis using the statistical software SPSS 22.0 (IBM Corp., Released 2013; IBM SPSS Statistics for Windows, Version 22.0. Armonk, NY, USA). Descriptive statistics were performed using median and range for continuous variables and frequency for categorical variables. Continuous variables were assessed for normality, while variables exhibiting a non-normal distribution underwent logarithmic transformation. Because many findings were equal to 0, logarithmic transformation was not possible for mHam; therefore, this variable was evaluated using nonparametric testing. One-way analysis of variance (with Bonferroni’s correction) or the Kruskal–Wallis test with Dunn’s multiple comparison test were used to compare groups. The comparison between three groups was conducted using Kendall’s tau test to analyze categorical data. Spearman’s rank correlation coefficients were employed to characterize the bivariate correlations. Multivariate analysis was performed using a logistic regression model. Analysis of specificity, sensitivity, and cutoff value was performed by creating the receiver operating characteristic (ROC) curve. The chosen level of statistical significance was established at 0.05. Data are presented as mean standard error of the mean and only significant *p*-values are shown.

## 5. Conclusions

Our study shows for the first time that suPAR and GDF-15 reflect endothelial injury in allo-HSCT recipients. In accordance with other patient populations, suPAR emerges as a marker of renal dysfunction, characterizing high risk in endothelial injury syndromes and in particular, HSCT-TMA. However, prior to their clinical usefulness, these biomarkers must undergo through rigorous validation in multiple cohorts. 

## Figures and Tables

**Figure 1 ijms-25-00231-f001:**
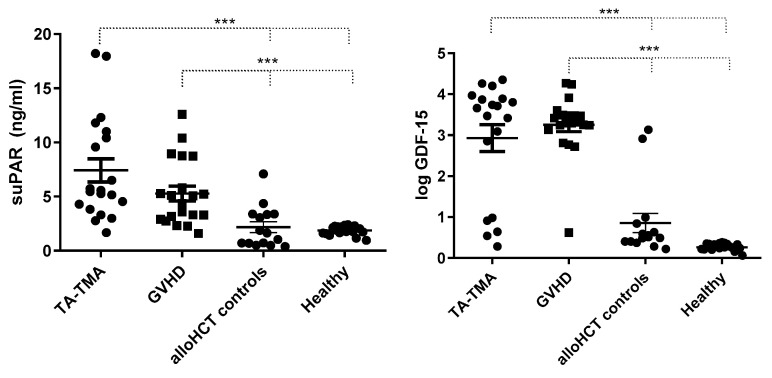
The levels of suPAR and GDF-15 were found to be significantly elevated in patients with HSCT-TMA and GVHD when compared to allo-HSCT recipients and healthy individuals (*p* < 0.001, Bonferroni’s correction). *** *p* < 0.001.

**Figure 2 ijms-25-00231-f002:**
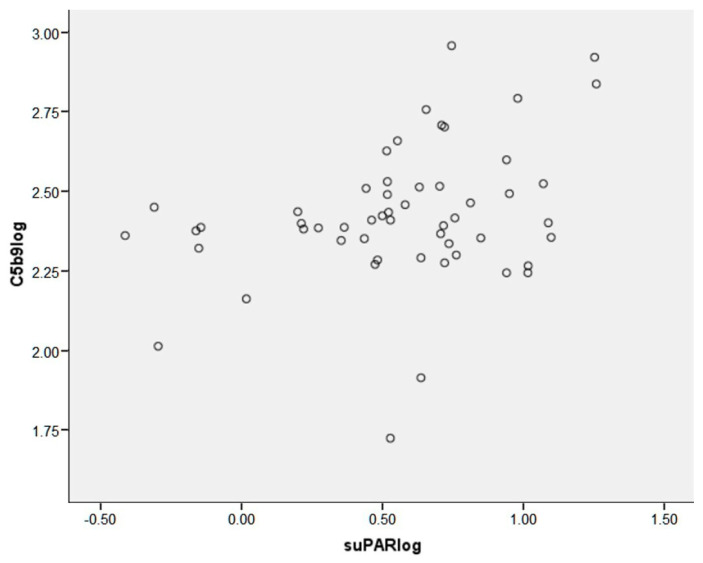
suPAR levels show a significant correlation with soluble C5b-9 levels (*p* = 0.013).

**Table 1 ijms-25-00231-t001:** Patient characteristics. Continuous variables are presented as median (range). The *p* value represents differences between the three groups, performed with the χ^2^ test for categorical and Kruskal–Wallis test for continuous variables. ALL = acute lymphocytic leukemia, AML = acute myeloid leukemia, CR = complete remission, EASIX = Endothelial Activation and Stress Index, GVHD = graft-versus-host disease, HLA = human leukocyte antigen, LDH = lactate dehydrogenase, TA-TMA = transplant-associated thrombotic microangiopathy.

	**TA-TMA (*n* = 20)**	**GVHD (*n* = 20)**	**Controls (*n* = 20)**	* **p** *
Age (year)	36 (17–56)	42 (19–52)	39 (18–49)	0.212
Disease type (n)				
AML	4	5	6	0.228
ALL	12	9	10	
Lymphoma	3	5	3	
Multiple myeloma	1	1	1	
Disease phase (n)				
Early CR	12	14	13	0.421
Late CR	4	3	4	
Relapsed/Refractory	4	3	3	
Myeloablative conditioning (n)	16	8	16	0.892
Donor (n)				
Sibling	8	10	9	0.732
Unrelated	8	5	6	
Haploidentical	4	5	4	
HLA-matched donor (n)	16	17	18	0.343
Follow-up (mo)	8.5 (2.7–102.1)	12.0 (2.9–32.2)	14.2 (4.5–79.1)	0.745
Infections (n)				
Bacterial	12	12	10	0.431
Viral	13	11	9	0.373
Fungal	10	7	12	0.653
GVHD (n)				
Severe acute	12	18	0	<0.001
Extensive chronic	17	19	0	<0.001
EASIX at day 0	1.57 (0.3–18.9)	1.37 (0.3–4.4)	1.4 (0.2–6.2)	0.565
EASIX at day 100	7.2 (1.1–118.2)	3.3 (1.8–12.1)	1.41 (0.2–30.8)	0.014
EASIX at last follow-up	22.7 (0.3–604.3)	7.8 (0.6–210.1)	0.89 (0.1–62.7)	0.001
Soluble C5b-9 (ng/mL)	325 (184–902)	243 (175–454)	227 (53–281)	0.001

## Data Availability

Data will be readily available upon request to the corresponding author.

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
