# Peer review of "Soluble Urokinase-Type Plasminogen Activator Receptor (suPAR) and Growth Differentiation Factor-15 (GDF-15) Levels Are Significantly Associated with Endothelial Injury Indices in Adult Allogeneic Hematopoietic Cell Transplantation Recipients"

_ijms, 2023, doi:10.3390/ijms25010231_

Round 1
Reviewer 1 Report
Comments and Suggestions for Authors
Authors assessed whether suPAR and GDF-15 reflect endothelial injury in allo-HSCT recipients.
This manuscript is potentially interesting, several issues arise.
Major points
1) This study is small size.
2) The onset date may be different between TMA and GVHA. Authors should show the sampling day.
3) Authors should show the usefulness of mearing these parameters.
4) This manuscript has low impact and needs addition of following points.
a) To compare suPAR and GDF-15 with vascular endothelial function.
b) To compare suPAR with fibrinolytic parameters
c) To compare GDF-15 with cell proliferation.
d) To explain the relationship between suPAR and GDF-15
e) To indicate relationship between these parameters and outcome.
Minor points
1) Figure 1 should be remade.
Reviewer 2 Report
Comments and Suggestions for Authors
The authors have made an interesting attempt at “Soluble Urokinase-Type Plasminogen Activator Receptor (suPAR) and Growth Differentiation Factor-15 (GDF-15) Levels Are Significantly Associated with Endothelial Injury Indices in Adult Allogeneic Hematopoietic Cell Transplantation Recipients”. The manuscript is interesting; however, the authors need to justify the scientific writing manuscript. Some of the general comments are provided below:
1. How was the selection process conducted for the different groups (HSCT-TMA patients, GvHD patients, control allo-HSCT recipients, and apparently healthy individuals)? Were there any specific criteria used for inclusion/exclusion, and how might those criteria influence the study's outcomes?
2. How were the results of the diagnostic markers (complement activation markers, EASIX, soluble C5b-9, suPAR, GDF-15) interpreted in the context of HSCT-TMA and GvHD diagnosis? Were there any unexpected correlations or trends observed?
3. The study included 20 patients in each category (HSCT-TMA, GVHD, control allo-HSCT, healthy controls). Do you think this sample size adequately represents the diversity within these patient populations, considering the potential variability in responses post-HSCT among different individuals?
4. The correlation between suPAR/GDF-15 and EASIX, LDH, creatinine, and platelet counts indicates a potential link between endothelial dysfunction and these biomarkers. How strong are these correlations, and how might they contribute to understanding the pathophysiology of HSCT-TMA and GVHD?
5. The association between suPAR and soluble C5b-9 levels specifically linked with thrombotic microangiopathy in HSCT-TMA is intriguing. How might this association impact diagnostic or prognostic approaches in identifying and managing HSCT-TMA?
6. The discussion delves into the pathogenesis of HSCT-TMA and GvHD, providing a comprehensive overview of endothelial dysfunction. How might these pathophysiological insights influence the development of targeted therapies or diagnostic approaches for these conditions?
7. While suPAR and GDF-15 are implicated in various diseases, their role in HSCT-related complications is relatively unexplored. How do the observed associations of these biomarkers with endothelial injury in HSCT patients align with their roles in other diseases? What unique aspects might their role in HSCT complications present?
8. The study acknowledges limitations, such as the small sample size and single-center design. How might these limitations impact the generalizability of the findings, and what should be the focus of future studies to validate and expand upon these results?
Round 2
Reviewer 1 Report
Comments and Suggestions for Authors
The onset date may be different between TMA and GVHA. Authors should show the sampling day. The sampling day after transplantation is better. Abstract needs attractive conclusion.
Author Response
Response to reviewer 1 points
The onset date may be different between TMA and GVHA. Authors should show the sampling day. The sampling day after transplantation is better.
Answer: Thank you for your query. Regarding HSCT-TMA sampling day was the 125th post-transplant day (rage 9-2931); whereas the sampling day for the GvHD was the 74th post-transplant day (range 12-141).
Abstract needs attractive conclusion.
Answer: Thank you very much for your valuable advice. We have accordingly revised our abstract. All the changes are marked in red color.
Reviewer 2 Report
Comments and Suggestions for Authors
The authors have addressed my queries and now the article is acceptable for publication.
Author Response
Dear reviewer,
We would like to thank you for taking the necessary time and effort to review our manuscript. We sincerely appreciate all your valuable and constructive comments, which helped us in improving the quality of the manuscript.
Sincerely,
Eleni Gavriilaki, MD PhD
Assist. Professor of Hematology
Medical School, AUTH